# Path of career planning and employment strategy based on deep learning in the information age

**Yichi Zhang** ⬥ *

Enrollment and Employment Division, Southwest Petroleum University, Chengdu, Sichuan, China

* 201899010062@swpu.edu.cn

## Abstract

With the improvement of education level and the expansion of higher education, more students can have the opportunities to obtain better education, and the pressure of employment competition is also increasing. How to improve students' employment competitiveness, comprehensive quality and the ability to explore paths for career planning and employment strategies has become a common concern in today's society. Under the background of today's informatization, the paths of career planning and employment strategies are becoming more and more informatized. The support of Internet is essential for obtaining more employment information. As a representative product of the information age, deep learning provides people with a better path. This paper conducts an in-depth study of the career planning and employment strategy paths based on deep learning in the information age. Research has shown that in the current information age, deep learning through career planning and employment strategy paths can help students solve the main problems they face in career planning education and better meet the needs of today's society. Career awareness increased by 35% and self-improvement by 15%. This indicated that in the information age, career planning and employment strategies based on deep learning are a way to conform to the trend of the times, which can better help college students improve their understanding, promote employment, and promote self-development. This study combines quantitative and qualitative methods, collects data through questionnaires, and uses deep learning model for analysis. Control group and experimental group were set up to evaluate the effect of career planning education. Descriptive statistics and correlation analysis were used to ensure the accuracy and reliability of the results.

## Introduction

In recent years, the employment prospects of college graduates have attracted widespread attention and discussions to the issue of reasonable future planning. First, the enrollment scale of universities is increasing year by year. The number of college students and talents is increasing, and higher education in China is becoming more and more popular [1]. Second, companies are still complaining about not finding the right talent in the face of an influx of college

**Data Availability Statement:** The original data has been uploaded to the submission system as a supplementary file.

**Competing interests:** The authors have declared that no competing interests exist.

graduates. There is a mismatch between Chinese universities and employers' needs for graduates, which has led to the problem of college graduates being rejected in the process of applying for jobs. Candidates who are competent for their jobs and have certain practical skills are hard to find. Meanwhile, college students who have already worked may again fall into a vicious circle of "job-hopping" once a year, which creates a psychological shadow of instability in the company. Faced with such problems as high mobility of college graduates and poor work ability, employers are becoming increasingly hesitant to recruit college students. With the improvement of economic and cultural level, the communication between countries is becoming increasingly close, and the demand for talents in various aspects such as comprehensive quality and practical application ability is also increasing. However, under the actual employment situation, the contradiction between "can't find a job" for college graduates and "find a talent" always exists. Therefore, when students are about to enter the society after graduation, a clear career planning and employment strategies are extremely important. In the current era of information explosion, how to obtain employment information and formulate career planning and employment strategies from a better path is a problem that needs to be faced. Due to its unique advantages, deep learning has attracted more scholars to explore it.

The basic principle of this article adopts deep learning as an important branch in the field of artificial intelligence, which has attracted much attention for its powerful data processing capabilities and automatic feature extraction capabilities. Its basic principle is to construct a multi-level neural network model and achieve modeling and understanding of complex data through multi-level nonlinear transformation. By continuously learning and adjusting network parameters, deep learning models can automatically learn the representation of data and play an important role in the fields of career planning and employment strategies. Although deep learning has shown amazing application potential in many fields, its application in career planning and employment strategies is still in the exploratory stage. Existing research shows that the traditional career planning education model has many problems, including students' insufficient career awareness and difficulty in obtaining employment information. Therefore, the core questions of this article are whether the career planning and employment strategy path based on deep learning can effectively improve college students' career cognitive level, and how to optimize the existing career planning education model.

The research structure is as follows. This article is divided into four parts: introduction, theoretical framework, empirical analysis and conclusion. In the theoretical framework section, the basic principles and applications of deep learning in career planning and employment strategies are introduced. In the empirical analysis part, questionnaire surveys and comparative analysis are used to verify the effectiveness and scientificity of career planning and employment strategy paths based on deep learning. Finally, the conclusion section summarizes the main findings of the study and proposes prospects for future research. This study aims to explore how deep learning technology can be applied in career planning and employment strategies to improve the accuracy, personalization, and adaptability of planning. Compared with traditional methods, deep learning has stronger data processing and pattern recognition capabilities, can better utilize big data resources, and provide individuals with more accurate career advice and employment guidance. Experiments showed that the path of career planning and employment strategies in the information age based on in-depth learning is more in line with the employment needs of contemporary people and has guiding significance for contemporary students, which is easier to be accepted and recognized by today's job seekers. In order to promote its promotion, the following aspects can be started from. The goal of the research is to explore the role of deep learning in career planning, evaluate its impact on college students, and provide a more scientific and effective guidance method for college students' career planning.

In order to eliminate conceptual confusion, this study specifically points out that 'Deeper Learning in Education' in the field of education and 'Deep Learning in AI' in the field of artificial intelligence are two completely different concepts. This study focuses on the latter, namely, the use of deep learning techniques in artificial intelligence to enhance students' career planning and employment strategy capabilities. The implementation of this technology and its application prospects in the field of education will be described in detail below.

## Literature review

For the path research of career planning and employment strategy in the information age, many scholars have carried out research on it at different levels and directions. Tavabie J A studied career planning for the non-clinical workforce, that is, opportunities to develop a sustainable workforce in primary care. He believed that such programs could support the recruitment and retention of an increasingly skilled workforce that moved between traditional health and social care providers, giving clinicians the opportunity to relieve themselves from onerous administrative tasks and support patients in an integrated care system [2]. The direction of Catanzano T's research was whether too much attention was paid to junior teachers and the career planning of middle and senior teachers was ignored. He focused more on the career planning of middle and senior teachers [3]. Branan J's main research was the construction of career planning courses for STEM PhDs. He showed that incorporating career development into doctoral programs could help biomedical students better understand themselves and the job market, and adopt an attitude of "they can do" when developing their own path [4]. Ziling mainly studied the career planning and education path research of aquatic animal medicine majors under the background of the "One Belt, One Road" initiative. Taking the students majoring in aquatic animal medicine in Guangdong Ocean University as an example, the actual level and existing problems of career planning for students in this major were expounded from five dimensions: career planning awareness, self-awareness, environmental awareness, goal planning, and implementation revision. Finally, he proposed solutions and reform proposals for education, teaching, and consulting paths from the perspectives of students, schools, families, government, and society [5]. The research of the above scholars clearly shows that there are limitations, which also indicates that many scholars still ignore the research on employment strategy, so the research direction is turned to deep learning. Lee J mainly studied the integration of digital twins and deep learning in cyber-physical systems and showed that the ultimate research goal of combining the two was towards physical manufacturing [6]. Kermany D S mainly studied the application of deep learning in medical diagnosis and disease identification which showed that it was feasible to use it in diseases such as blind retina [7]. Oshea T mainly studied the introduction of deep learning at the physical layer. He also proposed and discussed several novel applications of deep learning at the physical layer. Finally it was concluded the application of convolutional neural networks on raw IQ samples for modulation classification, achieving competitive accuracy with respect to traditional schemes that rely on expert features [8]. Schirrmeister R T focused on deep learning for electroencephalopathological decoding and visualization based on convolutional neural networks. He showed that convolutional neural networks could be applied to the task of distinguishing pathological and normal EEG recordings in a hospital EEG abnormal corpus. In decoding EEG pathology, the accuracy of using convolutional neural networks was much higher than the only published results for this dataset [9]. Through the research on deep learning, it is found that the research field and application scope of deep learning are extremely broad, which is mostly used in physics and medicine. However, few scholars have conducted in-depth research on the research fields used in career planning and employment, which

means that for deep learning, career planning and employment can be said to be an unfamiliar field. This requires people to carry out new development.

## Career planning and employment strategies in the information age of deep learning

In order to conduct in-depth research on career planning and employment strategies in the information age of deep learning, it is first necessary to understand the relevant overview. This section mainly introduces the general situation of career planning, employment strategy and deep learning in the information age.

In this study, it is crucial to explore the relationship between career planning, deep learning, and employment strategies. This relationship is bidirectional, that is, career planning and employment strategies may be affected by deep learning, and at the same time, career planning and employment strategies may also affect attitudes and applications of deep learning.

The impact of deep learning on career planning is manifested in the use of its data analysis capabilities to help individuals identify their strengths and career interests in order to provide customized career development guidance.

The application of deep learning in employment strategies enables enterprises to more accurately match talent needs and enhance the market competitiveness of job seekers by optimizing the recruitment process and employee training.

Career planning and employment strategies are interdependent, and together guide the decision-making of individuals on their career paths, ensuring the clarity of goals and the effectiveness of the path to realization.

By providing career planning guidance and organizing career development activities, the university builds a bridge between students and career opportunities and promotes the improvement of students' professional literacy.Provide internship and practical opportunities: Universities can work with industry to provide internship and practical opportunities for students, allowing them to gain experience in a real work environment and build professional networks. Establish relevant courses: Universities can offer courses related to career planning and employment strategies, including job market analysis, job search skills, interview preparation, etc., to help students master necessary skills and knowledge. Provide resource support: Resources such as university libraries, career centers and online platforms can provide students with job market information, career guidance tools and job search resources.

## Career and employment strategies in the information age

**Overview of career planning.** Career is all the behaviors and activities that a person engages in in life, including the attitude towards work, the value of work and so on. This is a continuous process, which can also be understood as a person doing different work in his life to achieve his ideals [10]. Simply, it is a person's work experience. Some scholars believe that career planning is to obtain and utilize relevant information, make corresponding plans and choices, and ultimately achieve one's career goals [11]. Some scholars regard it as an organic combination of the development of individuals and enterprises, and analyze and summarize the subjective and objective factors that affect their decision-making, thereby establishing a future development direction and a favorable cause, formulating corresponding action plans, and rationally arranging the work at each stage [12]. This concept explains that career planning is a systematic process from multiple levels, and thus reveals what characteristics an excellent career planning should have. From this point of view, it can be seen that before making career planning, one must first have a comprehensive understanding of himself, his career,

and the development of society, and then makes a suitable choice for his own development based on his own strengths.

**Inevitability of career planning.** At present, the employment pressure of college students is increasing. There are three main reasons. First, the expansion of university enrollment has resulted in a sharp increase in the number of university graduates. Second, the employment guidance agencies do not make corresponding adjustments, so that the effect of employment guidance is not obvious. Third, students do not know their own abilities and professional direction, nor do they have much confidence in their careers [13]. According to a survey conducted by Renmin University of China, currently, at the undergraduate level, only 17.6% of colleges and universities have started career counseling at the beginning of their freshman year, and after their senior year, only 52.9% of schools have started career counseling. If the employment counseling is carried out in the preparatory stage before employment, it may certainly not meet the actual needs of students. Due to the low educational level of college students' career planning and relatively little theoretical knowledge, it is difficult for them to play a greater role in career development.

Career planning needs to combine individuals and organizations, and formulate the most reasonable career goals according to individual interests, habits and abilities, and according to individual career characteristics. Judging from the current situation, the employment guidance work in some countries starts from primary school and has achieved good results. It can be seen that, taking career planning as the basis, giving comprehensive guidance to students, and fully considering the characteristics of students and the needs of society, can ensure the effect of guidance. From the perspective of college students, the impact of career planning on college students is obvious. It is an important condition and premise for students to achieve their employment goals, recognize themselves, and ultimately achieve better development.

**Problems existing in career standardization in the context of informatization.** At present, under the conditions of informatization, the career planning of college students is faced with two major problems: weak planning awareness and insufficient practical skills [14]. When college students choose careers, due to the influence of factors such as excess market supply, professional restrictions, personal hobbies, etc., they cannot make scientific and effective decisions. In addition, due to the existence of the "ivory tower" model, it is difficult for college students to obtain more social occupation information in the process of obtaining social occupations, and they fail to correctly understand the characteristics of their social occupations. As a result, they face difficult choices when choosing a career and lack social professional awareness. Most college students spend a long time with teachers and classmates, participate in less social practice activities, and have poor professional awareness. On the other hand, due to the ease of university life, college students have a sense of crisis in their work, and they do not pay enough attention to their work, which leads to their weak professional awareness. The development of their careers is only on the surface, and it is difficult to reflect rationality and competition. In work, only practice is the only criterion for testing truth and foothold. Therefore, it is necessary to organically link professional awareness and social practice to ensure its implementation and implementation. Although many college students have certain values and occupational orientations in their employment plans, due to the lack of a stable relationship with society, they have not integrated practical work into social development and lack practical ability, so that they can not adapt to the needs of society, making it difficult to realize their career in the society.

The connection between job hopping and deep learning can be analyzed through the following aspects. First, personalized matching: Deep learning technology can analyze personal skills, interests, work experience and other information, as well as the needs and cultural characteristics of different companies, thereby providing individuals with more accurate career

advice. When individuals consider changing jobs, deep learning can help them find a new position that best suits them, thereby increasing job satisfaction and career advancement. Second, market trend analysis: Deep learning can analyze trends and changes in the job market, including the development of different industries, the company's recruitment needs, etc. These analyses can help individuals understand the current employment environment, predict future development trends, and provide reference for job-hopping decisions. Third, emotional analysis: Deep learning technology can also analyze an individual's emotional state at work, including job satisfaction, stress feelings, etc.Fourth, intelligent recommendation: An intelligent recommendation system based on deep learning can recommend suitable job opportunities and company information to individuals. When individuals are considering changing jobs, these recommendations can help them quickly find positions of interest and learn about relevant companies, allowing them to make more informed decisions. Through the application of deep learning, individuals can have a clearer understanding of their career development path, find the job opportunities that best suit them, and achieve their career goals.

**Employment strategy.**   Employment strategies refer to the specific measures and plans adopted by individuals or organizations to achieve employment goals. In career planning, employment strategies refer to the specific action plans and methods developed to achieve these goals after career goals are determined. Employment strategies cover various strategies and methods adopted by individuals during the job search process, including but not limited to: finding suitable job opportunities, writing resumes and cover letters, participating in interviews, establishing professional contacts, improving professional skills, etc.

Employment strategy is of great significance in career planning, which is mainly reflected in the following aspects: first, achieving career goals. The ultimate goal of career planning is to realize personal career vision and goals. Employment strategies are specific ways and methods to achieve these goals. Effective employment strategy plays a decisive role in achieving career goals, enhancing market competitiveness, adapting to changes in career environment and enhancing employment satisfaction, and is an indispensable part of personal career development. It is a key step in achieving personal career goals and an important guarantee for personal success in the job market.

From an individual's point of view, employment strategy is achieved through learning, training, and cultivating one's own career selection concept, professional ethics, professional quality, professional skills, professional skills, knowledge, career selection, and job search [15]. Developing the right employment strategy helps students to have a correct view of their careers. A proven employment strategy can help one develop the right job skills as well as save time for more meaningful work with limited time. To find a suitable employment strategy, one needs to fully understand and master his work. In this information age, the importance of information is unquestionable, and one must have enough time to obtain more information.

**Using career planning to promote employment strategies in the information age.**   In order to improve the ability of college students to formulate employment strategies, it should start from enriching the forms of employment guidance, standardizing the content of employment guidance, and improving the career planning system. Traditional employment guidance only focuses on employment services, emphasizes employment policies and interview methods, and emphasizes the cultivation of employment concepts. Therefore, it is necessary to establish a more systematic career planning and employment strategy system[16]. Vocational guidance is an important part of college students' psychological counseling. It mainly aims at the problems encountered in the planning process, and proposes corresponding countermeasures and skills. Meanwhile, it is necessary to combine with modern technology and establish a website that is suitable for it. On the one hand, it can provide more employment opportunities for college students. On the other hand, it provides a communication platform for college

students, so that students can enrich their employment information and achieve their career development goals through the network according to their own needs. Due to the constraints of traditional employment policies, talent training and career planning remain on the surface, which cannot fully reflect the teaching goal of "student-oriented, all-round development", and cannot ensure the effect of employment guidance. Therefore, on the premise of in-depth understanding of career planning, it is necessary to increase investment in relevant courses, so that students can be familiar with relevant theoretical knowledge as much as possible, so as to solve various problems in practical operation. Career planning can be improved by constructing a career education system, establishing a scientific evaluation system, and strengthening professional talent cultivation, so that college students can have a clearer understanding of their career planning and a basis can be provided for formulating their employment strategies.

In today's society, the slowdown of economic growth, social transformation, and people's expectations for employment are constantly increasing, which makes the employment of college students face new situations and challenges[17]. With the advent of the era of big data, the emergence of artificial intelligence, the reduction of network costs, and the popularization of smart phones, people's utilization and dependence on information resources are constantly improving, and the demand for information resources is also increasing. As a product of computers and artificial intelligence, deep learning is more suitable for the needs of today's society for career planning and employment strategies[18]. Therefore, the research direction is turned to deep learning.

## Deep learning

**Overview of deep learning.** This article focuses on deep learning in depth. Deep learning is a machine learning technology inspired by the neural network structure of the human brain [19,20]. It learns and understands data through multi-level neural networks to achieve efficient processing and accurate prediction of complex problems. Deep learning technology has automatic learning characteristics. It can extract abstract features from large-scale data and continuously optimize model performance during the model training process. Although traditional career planning and employment strategy models have existed for many years, with the rapid development of society and the advancement of technology, these models have become lagging behind and limited. Traditional models are often based on static data and rules and lack adaptability to complex changing environments.

Neural networks and their related techniques constitute deep learning. Deep learning is a function with good fitting performance, which can make the output approximate to the actual sampled value by adjusting the parameters in the function [21]. The model contains multiple levels, and the input of each level is the external output, so as to describe the data hierarchically. For deep learning, its learning task has a specific purpose. Process is learning the necessary characteristics in order to achieve a purpose. It exploits domain knowledge to a minimal extent, and obtains information directly from the initial input.

Deep learning was first used in speech recognition, and it has been widely used in natural language processing, computer vision, reinforcement learning, etc. [22]. Then, super-resolution analysis is performed using a convolutional neural network. On this basis, a content-based language model is established using a recurrent neural network. Based on this, a deep reinforcement-based method is proposed to realize the duration control of the robot. Although the interpretability research of deep learning still has many deficiencies, it has greater practical value through in-depth research on its structure and optimization methods. In deep learning, fully connected hierarchies were first used. The fully connected layer structure is currently the most widely used one, which makes the input layer and the output layer completely connected.

Since such neural networks have a large number of parameters, they cannot be widely used in deep learning. This structure is widely used in the last and second layers of neural networks. It establishes the relationship between the hidden layer and the output as the output layer of reinforcement learning. Neural network models are rarely used because the weight of the neural network in the entire connection layer is too large to be trained. Convolutional layers are a structure widely used in modern neural networks. The convolutional layer is composed of a set of spatial filters with adaptive parameters, and its output is a set of image sequences filtered from two-dimensional spatial data. The concept of convolutional layers is triggered by the local sensory areas of the human eye, and each part of the human eye in the image space is received by a set of receptors. In different parts, the same receptors are present. This invariance is accomplished through a weighted distribution. That is, in the convolutional layer of the neural network, the network connection weights of the input end and the output end of the rectangular area are equal. A network that combines a convolutional layer with a general neural network is called a convolutional neural network, or CNN [23].

Recurrent neural network [24,25] (also known as RNN) can solve the problem of processing continuous tasks well with traditional feedforward neural networks. Generally, a neural network without feedback is called a forward neural network. The recurrent neural network includes not only the feedback from the input to the hidden layer, but also the feedback from the hidden layer to the input and the output to the hidden layer. Recurrent neural networks are often used to build sequence structures and sequence models, and are widely used in speech recognition, machine translation, and natural language processing. Because the traditional recurrent neural network can no longer be used, a variant, long and short-term memory network, has been produced, which also has been widely used. The long short-term memory network is an improved RNN, which adopts a new technology based on RNN to realize the selective control of continuous data based on RNN [26,27]. Meanwhile, the long-short-term memory network can effectively overcome the gradient explosion and gradient disappearance of traditional RNN in reverse transmission.

While improving the neural network model, the optimization algorithm has also been further developed. The most commonly used deep learning algorithms today are based on back-propagation. In deep learning, the most common and effective method for parameter update is to use the back-propagation (BP) method, and then use the heuristic optimization method to update the parameter gradient. The key to the BP algorithm is to regard the chain solution process as the reverse transmission of errors in the neural network. The basic principle of the BP algorithm is very simple. First, the objective function is obtained, and then the chain law is used to solve each parameter. The currently widely used deep learning frameworks are based on computational graphs and BP, realizing self-optimization.

Scholars such as Guleria P study how deep learning can help with career planning [28,29]. Deep learning provides powerful support and assistance for personal career planning through personalized recommendations, skills matching, and market trend predictions. Career planning methods based on deep learning can help individuals better understand the career market, improve employment competitiveness, and achieve personal career goals and development. By analyzing personal career characteristics and market trends, deep learning technology provides personalized career recommendation, skill matching analysis, and market development trend prediction to help individuals develop and optimize career planning, enhance employment competitiveness and market adaptability.

Deep learning models analyze multiple types of data, such as text, images and video, to provide comprehensive career development information and help students understand the current situation and trends in the industry.

Deep learning models use their advantages in pattern recognition and prediction to reveal complex relationships between data and provide accurate prediction and decision support for career planning and employment strategies.

**Deep learning mode.** Deep learning can provide students with personalized career planning services through personalized advice, skill matching, market prediction and intelligent recommendation.

This article chooses the deep learning model as the experimental model for the following reasons:

1. Deep learning models have powerful data modeling capabilities and can handle large-scale and high-dimensional data, which is consistent with the data analysis capabilities required for career planning and employment strategy research [30,31].

2. Deep learning models perform well in information extraction and pattern recognition, and can discover potential patterns and associations in data, providing researchers with deeper insights [32,33].

3. With the development of the information age, the amount of data continues to increase, and the complexity of the data is also increasing. Traditional analysis methods may not be able to handle this data well, and deep learning models can better cope with this challenge.

Choosing deep learning models as research tools is of great significance for improving research data analysis capabilities, discovering hidden patterns in data, achieving more accurate predictions and classifications, and providing students with better career planning and employment strategy suggestions. The application of deep learning models makes research more cutting-edge and convincing, and helps to more comprehensively understand and solve problems in the field of career planning and employment strategies.

Before introducing deep learning models, a general introduction to neural networks is given first. A neural network is a feedforward neural network composed of neurons. Its basic neuron structure is shown in Fig 1:

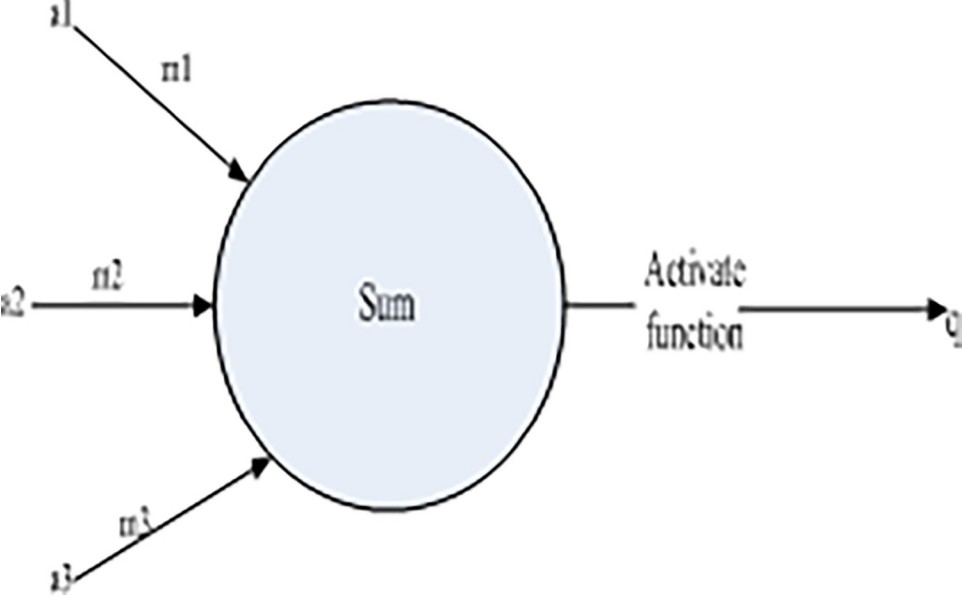

**Fig 1. Neural network of artificial neural network.**

As shown in Fig 1, the information received by the neuron goes through the weight and operation of the neuron, which then is sent to the activation function, and then the operation is performed. Finally the output of the neuron is obtained.

a—Input vector;

m—Neuron weight vector;

y(a)-Activation function, which is used to introduce nonlinear properties;

q—Final output of the neuron.

Their calculation relationship satisfies Eqs 1 and 2:

$$q = y(am) \tag{1}$$

$$q = y(a_1 m_1 + a_2 m_2 + a_3 m_3) \tag{2}$$

In deep learning, weighting, initialization weighting and other methods are used to reduce the difficulty of training and make it have typical hierarchical depth characteristics. Convolutional neural networks and recurrent neural networks are currently the two most widely used types. Their variant long and short-term memory network has better performance than traditional convolutional neural network.

(1) Convolutional neural network

Convolutional neural networks are feedforward neural networks, which can be thought of as sharing weights into an ordinary fully connected network. The algorithm is similar to a general neural network and consists of neural units with learnable weights and biases. Each node, the neuron, accepts the product of vectors.

Convolutional neural networks have different levels of neuron features. The traditional neural network considers the number of neurons at all levels, and the calculation from the middle layer to the next layer is realized by a matrix product. Convolutional neural networks need to comprehensively consider factors such as the width, height, number of channels, and number of filters of the convolutional layer. A convolutional layer in a convolutional neural network can be thought of as a set of fixed number of filters. Assuming that the input is an image, the number of channels of each filter is the same as the number of channels of the input image. The convolutional layer uses multiple filters to filter the image. The number of filtered images is then equal to the number of filters in the convolutional layer. The specific structure is shown in Fig 2:

(2) Recurrent neural network

A recurrent neural network is a type of neural network that utilizes serialized information. In traditional neural networks, inputs and outputs are assumed to be independent of each

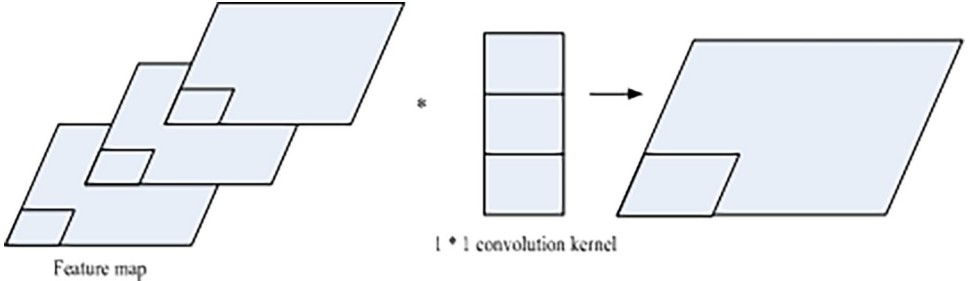

Feature map

**Fig 2. Schematic diagram of convolution layer.**

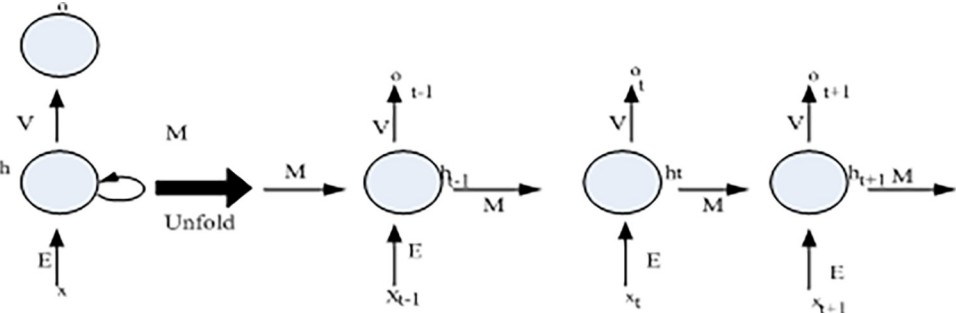

**Fig 3. Cyclic neural network and its deployment diagram.**

other. The recurrent neural network has a certain memory ability and performs the same operation on each unit in the sequence. The output is relative to the previous input, where each element in the sequence corresponds to a point in time. The structure of the recurrent neural network can be expanded into the form of a feedforward neural network according to time, as shown in Fig 3:

It can be seen from Fig 3 that

$h_t$—State of the hidden layer at all times;

$o_t$—Time output;

E, V, M—Weights of the neural network.

From this, the equation satisfied by the state of the hidden layer at different times can be known from Eq 3:

$$h_t = y(E_t + M_{h_{t-1}}) \tag{3}$$

In Eq 3, y is a commonly used nonlinear function, such as a hyperbolic sine function, a rectified linear unit, etc.

t—Output that can be viewed as a probability vector.

$$o_t = softmax(V_{h_t}) \tag{4}$$

Recurrent neural networks can generally be replaced by long and short-term memory networks to process longer time series.

The long-short-term memory model (also called LSTM) is a temporal recurrent neural network with a special structure, which is suitable for processing and predicting time series with large delays. At present, RNN-based long and short-term memory models are widely used in one-dimensional deep neural networks [34]. It can be seen from the back-conduction equation of RNN and the back-propagation algorithm of MLP that when the residual is back-propagated in the time dimension, it decays substantially over time.

$$\text{RNN}: \ \sigma_a^t = y'(b_a^t)\left(\sum\nolimits_j \sigma_j^t \gamma_{aj} + \sum\nolimits_{a'} \sigma_{a'}^{t+1} \gamma aa'\right) \tag{5}$$

$$\text{MLP}: \ \sigma_a = y'(b_a) \sum\nolimits_{a'=1}^{a_{\varsigma+1}} \gamma_{aa'} \sigma_{a'} \tag{6}$$

Since the slope of the activation function y changes less outside the interval, that is, the reciprocal value of the function y is small, the value of the activation function decays after a period of time. As the accumulation increases, the final gradient decays gradually becomes larger, so that the feedback signal at a certain time has basically no effect on the longer-term

information.

$$\frac{\partial Q}{\partial \gamma_{cd}} = \frac{\partial Q}{\partial b_d} \frac{\partial b_d}{\partial \gamma_{cd}} = \sigma_d g_c \tag{7}$$

The LSTM network replaces the hidden layer units in the RNN with storage blocks, and uses memory units to store data, which is expressed as Eq 8:

$$h^t = m^t \odot c^t + h^{t-1} \tag{8}$$

$m^t$—Time input.

The output gate is similar to producing a 0–1 vector to control the output of the memory cell to the output layer, as shown in Eq 9:

$$v^t = h^t \odot q^t \tag{9}$$

To enhance the processing power of LSTM, forget gates are introduced to replace edges with weight 1. The forget gate forgets the previous content by multiplying the content in the memory cell with the forget gate element-wise. The forgetting gate is the same as the input/output gate. They all accept it as input. The update equation of the current LSTM memory unit is as Eq 10:

$$h^t = m^t \odot c^t + y^t \odot h^{t-1} \tag{10}$$

Among them,

$b_d^t$—Input of unit d at time t;

$k_d^t = y(b_d^t)$—Non-linear mapping of the input of element d;

$\zeta, \varphi, \gamma$—Input gate, forget gate, output gate;

R—Number of memory cells;

$h_r^t$—State of memory unit r at time t;

y—Activation function of the gate;

m, a—Activation functions of the input and output of the memory unit, respectively;

C—Input layer size;

A—Size of the hidden layer memory unit;

T—Output layer size.

The forget gate: In the memory block of LSTM, only the output of the memory unit at the previous moment is transmitted to this unit. Other data such as memory cell gates or memory cell inputs are only visible inside the cell. The forget gate is used to control the memory cell gate at the last moment, namely, as shown in Eqs 11 and 12:

$$b_\varphi^t = \sum_c \gamma_{c\varphi} x_c^t + \sum_a \gamma_{a\varphi} k_a^{t-1} + \sum_r \gamma_{r\varphi} h_r^{t-1} \tag{11}$$

$$k_\varphi^t = y(b_\varphi^t) \tag{12}$$

The input gate: This gate controls the input of the memory cell at the current moment, as shown in Eqs 13 and 14:

$$b_\varsigma^t = \sum_c \gamma_{c\varsigma} x_c^t + \sum_a \gamma_{a\tau} k_a^{t-1} + \sum_r \gamma_{r\varsigma} h_r^{t-1} \tag{13}$$

$$k_\varsigma^t = y(b_\varsigma^t) \tag{14}$$

The output gate: This gate controls the memory output, as shown in Eqs 15 and 16:

$$b_\gamma^t = \sum_c \gamma_{c\gamma} x_c^t + \sum_a \gamma_{a\gamma} k_a^{t-1} + \sum_r \gamma_{r\gamma} h_r^t \tag{15}$$

$$k_\gamma^t = y(b_\gamma^t) \tag{16}$$

The equation for calculating the output of the memory unit is as Eq 17:

$$k_r^t = k_\gamma^t a(h_r^t) \tag{17}$$

This section mainly introduces the deep learning network model and deep learning-related algorithms used for the verification of the method in this paper, which can provide relevant theoretical foundations for subsequent experiments.

Compared with existing online employment platforms, the website proposed in the study can adopt strategies to differentiate itself. The website can use deep learning technology to conduct a more precise analysis of users' interests, skills, and experiences and provide personalized job recommendations. Compared with other platforms, the recommendations of this website are more targeted to the needs of users, improving the matching degree. The website can focus on a specific industry or field and provide more professional and in-depth employment services. Work closely with industry partners to provide up-to-date employment information and increase user attention.

In view of the advantages of using deep learning technology for personalized career recommendation proposed in this study, this paper further discusses the specific application and potential impact of this technology on career planning path construction. Based on the fact that websites can provide more accurate analysis of users' career interests, skills, and experiences, this paper proposes the following hypotheses to evaluate the effectiveness of deep learning models in improving the cognitive depth and accuracy of college students' career development. These assumptions are not only based on the description of the technical functions of the website, but also rooted in a deep understanding of the needs of students in the education information age.

H1: Career planning path constructed by deep learning model can effectively improve the depth and accuracy of college students' cognition of career development.

H2: Deep learning technologies optimize students' access to career information, provide more comprehensive and accurate market insights, and strengthen the basis for career planning.

H3: The career planning path supported by deep learning promotes students' self-development and adaptability, and helps them adapt to changes in society and the workplace.

## Application of deep learning technology in education

In this study, deep learning techniques are applied to the educational field of career planning and employment strategies. The technology relies on multi-layer neural network models, including CNN and LSTM, to enable in-depth analysis of educational data. The CNN structure captures local features of image and text data through its multilayer filters, while the LSTM network optimizes the processing of sequence data to learn and predict long-term dependencies in time series. These models provide students with personalized career development advice and employment strategies by training complex patterns in learning data. In the study, the application of deep learning technology not only improves students' cognition level of career planning, but also optimizes their access to employment information, and promotes self-development and adaptability. Through empirical analysis, this study validates the

effectiveness of deep learning application in the field of education and demonstrates its potential in enhancing students' career competitiveness.

## Experiment design and validation

In order to verify the effectiveness and scientificity of the information age career planning and employment strategy paths based on deep learning mode, and to help more students establish the correct career outlook, it is necessary to select a scientific information dissemination path, formulate the correct employment strategy, and maintain the stability of the employment market. This study collected and surveyed employment information of graduates in the job market. Through a questionnaire survey, the path of career planning based on deep learning and the path of traditional career planning and employment strategies on the changes of students' cognition and themselves were compared to draw conclusions. Research showed that the path of career planning and employment strategy based on deep learning in the information age can better help unemployed people understand career planning and formulate employment strategies to finally choose a correct job [35].

This paper, mainly through the understanding and analysis of the teaching guiding theory of students' career planning, and through interviews and investigations on the problems of the path of students' career planning and employment strategies, found that the factors that affect students' career planning and employment strategy formulation mainly included the following two aspects. In colleges and universities, due to the limited scope of teaching and the poor performance, the degree of networked teaching of professional life planning was low. Due to their own factors, schools were not familiar enough with the teaching of students' career planning and were not clear about the methods of career planning.

The following steps are used for data analysis in the study.

Data cleaning: deal with missing and outliers to ensure accurate and complete data. Descriptive statistical analysis of sample characteristics. Correlation analysis explores the relationship between variables.

Descriptive statistical analysis: Descriptive statistical analysis is a simple but effective method for understanding the basic characteristics of a sample.

Calculation of the mean, median, and mode: They are used to understand the central tendency of the data.

Calculation of standard deviation and variance: They are used to understand the degree of dispersion of data.

Correlation analysis: Correlation analysis is used to explore the relationship between two or more variables. In this article, the Pearson correlation coefficient is used to measure the degree of linear correlation between two variables. This analysis helps researchers understand the correlation between different variables, such as whether there is a correlation between occupational cognitive level and educational background.

This statistical approach to this experiment is sufficient to address the complexity of the problem addressed for the following reasons:

1. The main purpose of this study is to compare the impact of deep learning-based career planning paths and traditional career planning paths on college students' career cognitive level and employment strategy acquisition. Although the context of the problem may be complex, the core question of the study is the impact on students by comparing different educational pathways. Therefore descriptive statistical analysis and correlation analysis can provide sufficient information to understand these effects and draw conclusions.

2. This study involves a survey of 600 students, 300 from each school, and the sample size is relatively large. A large enough sample size can increase the credibility of statistical results

and reduce the impact of chance factors on the results. In this case, simple statistical methods are sufficient to meet the needs of the research.

3. The research question is mainly to evaluate the impact of different educational paths on students' career cognitive level and employment strategy acquisition. The evaluation of these aspects can be achieved through simple statistical methods, such as mean comparison, frequency distribution, etc. Therefore, overly complex statistical analysis methods are not required to solve these problems.

4. The descriptive statistical analysis and correlation analysis methods used in this article can help understand students' career planning and employment strategy acquisition, and compare them with different education paths. Within the scope of the study, these simple statistical methods are sufficient to support the research purpose, and further complex analysis is not necessary.

For statistical testing, through statistical hypothesis testing, appropriate statistical hypothesis testing methods, t-tests are used to test whether the differences between different groups are statistically significant. Specifically, the differences between the traditional path group and the deep learning-based path group in terms of career cognitive level and employment strategy acquisition is compared.

## Experimental design

In order to verify the traditional career planning and employment strategy path and the path based on deep learning method, experiments were carried out to solve the above problems. Two local colleges and universities were selected for the experiment. The graduate path of school A was a traditional career planning and employment strategy and school B was a path based on deep learning. There were 600 students in the experiment, 300 from each school including 150 boys and girls. The two schools were not much different in size. Students in the two schools were educated in different paths, and they were tested after one semester. The questionnaire contained about 20 questions, covering students' awareness of career planning and employment strategies, educational background, career preferences, etc. The overall number of responses was 308 complete responses, which were distributed through paper questionnaires and participants responded anonymously.

The reason why experimental design was chosen as the research method in this study was mainly to compare the impact of deep learning-based career planning and employment strategy paths with traditional paths on the occupational cognitive level and employment strategy acquisition of college students under controllable conditions. The influence of other variables could be controlled through experimental design, making the experimental results more credible and comparable.

The experimental process operation was divided into two stages. First, before the experiment started, a questionnaire was conducted on the students participating in the experiment to understand their basic situation and career planning awareness. Then during one semester, the two groups of students received different education on career planning and employment strategies. After the education, a questionnaire was administered to the students again to compare the differences in career awareness levels and employment strategy acquisition between the two groups of students. During the experiment, the interference of other influencing factors was controlled as much as possible, such as students' basic knowledge level, school background, etc. At the same time, the experimental process was randomized to ensure the credibility and reliability of the experimental results.

The data analysis used statistical analysis methods to analyze the experimental results, compare the differences in career awareness levels and employment strategy acquisition between the two groups of students, and verify the proposed hypotheses.

In the study, descriptive statistics were used to analyze the data collected through the questionnaire survey. When selecting interviewees, consideration should be given to the purpose and questions of the study. College students were selected as respondents in this study because they were the main targets of career planning and employment strategies. Respondents could be selected through random sampling or stratified sampling to ensure a representative sample. In addition, respondents with different backgrounds and experiences could be selected according to the characteristics of the research question to obtain richer information.

## Experimental deconstruction

**Results of limited coverage of on-campus courses and poor teaching effect.**   In this paper, students from two schools were surveyed to explore their main sources of knowledge about career planning and employment strategies. The questionnaire was designed through literature review and expert consultation, and 600 students were selected by random sampling to ensure the representativeness of the sample. The data collection followed ethical standards, including anonymity and informed consent. Descriptive statistical methods were used in statistical analysis to classify "yes" and "no" answers, and to present the proportion of dependence on school education and other information sources in percentage form.

Regarding the limited coverage of the on-campus courses and the poor teaching effect, the students of the two schools were asked "the main source of knowledge about your career planning and employment strategy" to answer. The specific results are shown in Table 1:

It can be seen from Table 1 that for the source of career planning knowledge, only 40% of the students based on the traditional path relied on the school, and nearly 80% of the students based on the deep learning path relied on the school. This showed that the method based on deep learning can meet the needs of school students to understand the knowledge of career planning and employment strategies.

## Results of the network problem of career planning and education

Regarding the problem of networked career planning education, when the knowledge about career planning of the school's recruitment website was sufficient, the specific understanding results are shown in Fig 4:

As can be seen from Fig 4, students in schools A and B differed greatly in their choice of content satisfaction with regard to employment on the school website. Most of the students in school A were not satisfied, while the students in school B were mostly satisfied. The specific comparative satisfaction chart is shown in Table 2:

It can be seen from Table 2 that in the satisfaction answer to the column of "whether employment knowledge can be satisfied on the school website", the answers of school students based on the deep learning path were "basically enough". This meant that knowledge storage and publicity in school careers and employment strategies were in place through deep learning, which was sufficient for students to learn and obtain information. The conclusion here confirmed hypothesis H2. Career planning and employment strategy paths based on deep

**Table 1. Comparison of students' knowledge sources of career planning.**

| option | A | | B | |
| --- | --- | --- | --- | --- |
| | number of students | Percentage (%) | number of students | Percentage (%) |
| yes | 120 | 40% | 212 | 70.66% |
| no | 180 | 60% | 88 | 29.33% |
| total | 300 | 100% | 300 | 100% |

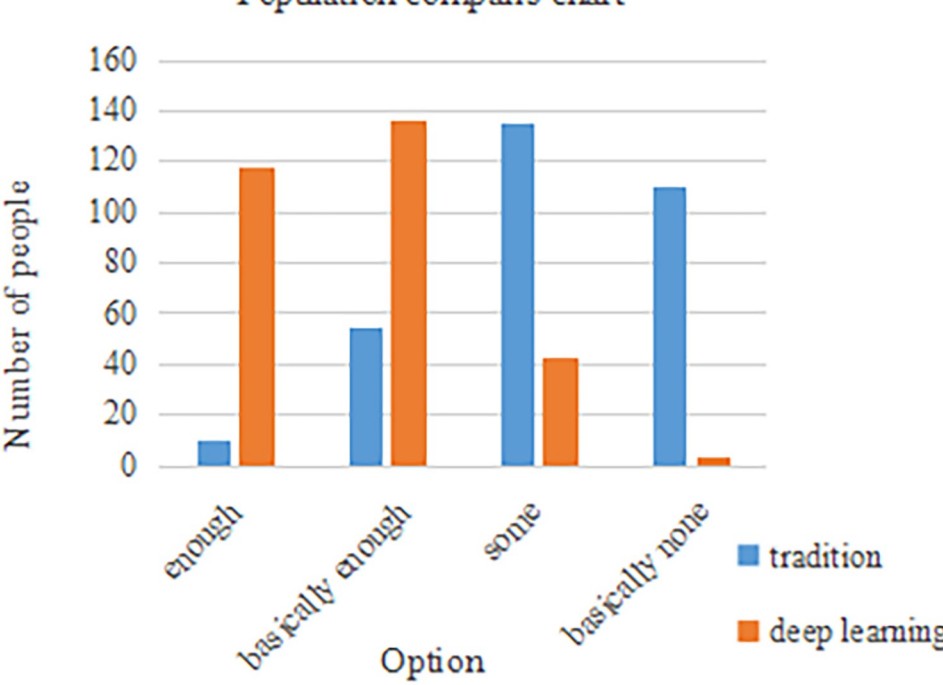

**Fig 4. Students' satisfaction with college website employment knowledge.**

learning could optimize students' access to employment information. Deep learning technology could provide students with broader and more accurate employment information, thereby helping them better understand the needs and trends of the job market and providing stronger support for their career planning.

## Results of career planning

Regarding the understanding of career planning and employment strategies, the specific survey results are shown in Table 3:

When it came to whether it is necessary to carry out career planning and employment strategy formulation, the specific results are shown in Fig 5:

As can be seen from Fig 5 above, in terms of understanding of career planning and employment strategies, the feedback from the two questions asked showed that students from school A were far less knowledgeable about careers than students from school B. Therefore, on the question of whether it was necessary, most of the students in school B answered yes, but only a small number of students in school A thought it was necessary. It can be seen that, compared

**Table 2. Comparison of students' satisfaction with school network.**

| | A | B |
|---|---|---|
| option | percentage (%) | percentage (%) |
| enough | 3 | 39.3 |
| basically enough | 18.3 | 45.3 |
| some, but not enough | 45 | 18 |
| hardly any | 36.7 | 1 |
| total | 100 | 100 |

**Table 3. Comparison of understanding of career planning and employment strategies.**

| option | A | | B | |
|---|---|---|---|---|
| | number of students | percentage (%) | number of students | percentage (%) |
| very familiar | 18 | 9 | 127 | 42.3 |
| general understanding | 82 | 27.3 | 135 | 45 |
| I don't know much | 147 | 49 | 30 | 10 |
| do not understand | 53 | 17.7 | 8 | 2.7 |
| total | 300 | 100 | 300 | 100 |

with the path conditions of deep learning, the traditional path method made most students lack the knowledge of career planning and employment strategy. The conclusion here confirmed hypothesis H1. Career planning and employment strategy paths based on deep learning could significantly improve college students' career awareness. Career planning and employment strategy paths constructed through deep learning technology could better help students understand the current status and trends of career development, thereby improving their understanding of career planning.

### Results of problems of unclear career planning methods

The result analysis of the problem of unclear career planning methods is shown in Fig 6:

As can be seen from Fig 6, most of the traditional path students did not understand the method of career planning and employment strategy formulation, but students based on deep learning methods had a relatively deep understanding of this. This showed that the rich

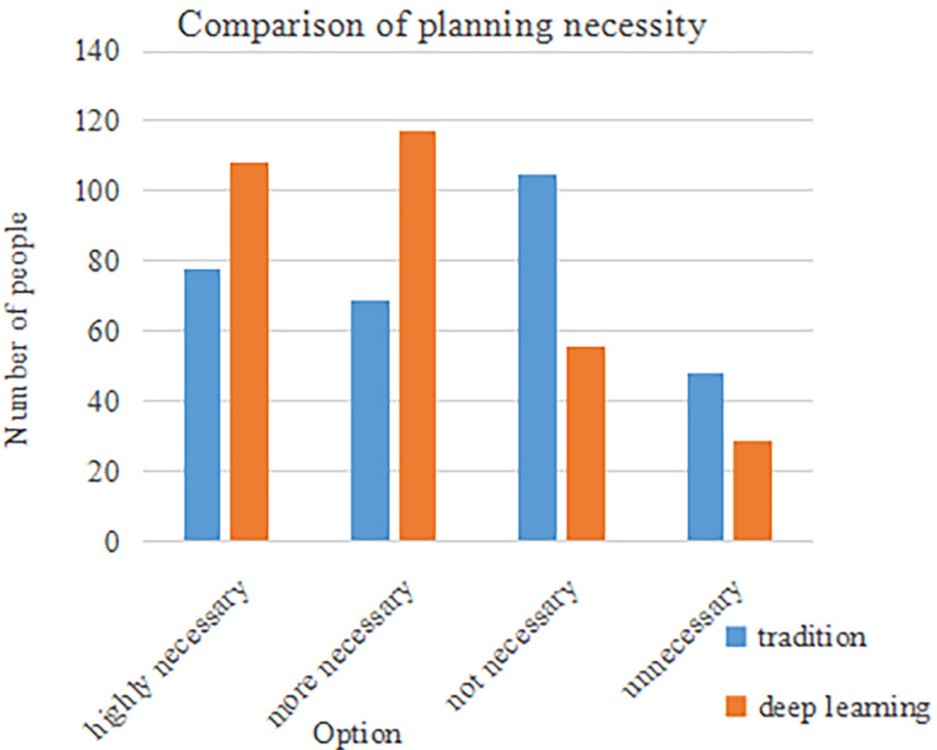

**Fig 5. Cognitive comparison between career planning and employment strategy formulation.**

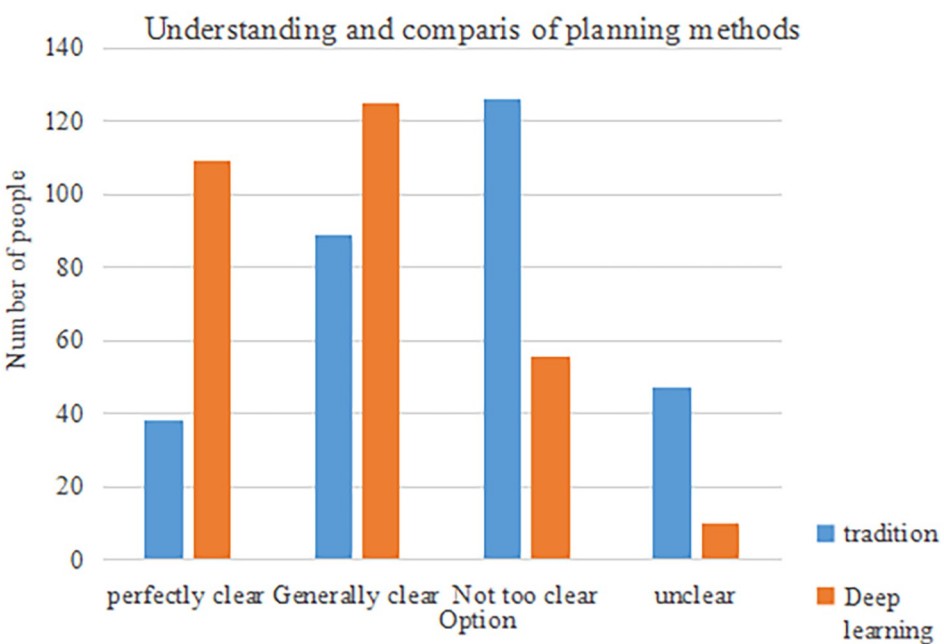

**Fig 6. Comparison of understanding of career planning methods.**

information through the deep learning path can fully meet the needs of students in the aspects of informatization career planning and employment strategy formulation. Students who had gone through this method had a deeper understanding of this issue, which could better help students formulate strategies and choose careers. Through the above comparison, finally, the students' overall cognition of career planning and employment strategies through two different paths and the changes of students' own quality were obtained. The details are shown in Table 4:

As can be seen from Table 4, through experimental analysis, it was found that students had a deeper understanding of the issues of career planning and employment strategies based on deep learning informatization than the traditional path. In general, compared with the traditional path method, the path based on in-depth learning could promote students' awareness of career planning and employment strategies by about 35% in college education. In terms of self-improvement, the overall improvement was increased by 15%. This showed that the career planning and employment strategy path based on deep learning in the information age can help students understand themselves, gain knowledge in this aspect, and formulate correct planning and employment strategies. The conclusion here confirmed hypothesis H3. Career planning and employment strategy paths based on deep learning could promote students' self-development and adaptability improvement. Career planning and employment strategy paths constructed through deep learning technology could stimulate students' self-awareness and development potential, helping them better adapt to social and occupational changes.

**Table 4. Final changes of the two paths of students.**

| option | A | | B | |
|---|---|---|---|---|
| | number of people | percentage (%) | number of people | percentage (%) |
| cognitive change | 124 | 41.3 | 229 | 76.3 |
| self change | 149 | 49.7 | 194 | 64.7 |

## Development of hypotheses

According to the three hypotheses of this article, the hypotheses have been gradually verified in the results. The hypothesis of H1 can be verified in the "Career Planning Results" section of the experimental results. By comparing the effects of two different paths on students' career cognitive levels, conclusions can be drawn. Experiments have shown that the career planning path based on deep learning enabled students to have a deeper understanding of career planning. Therefore, this part of the experimental results can verify H1.

Reason: Paths based on deep learning enable students to have a deeper understanding of career planning, because deep learning technology can provide broader and more accurate career information and help students understand the current status and trends of career development [36]. Gap: There are still some students who do not fully grasp the career information provided by deep learning. There are difficulties in applying this information to actual career planning. Although this can lead to an improvement in cognitive level, not every student can achieve it, and the degree of improvement varies.

The hypothesis of H2 can be verified in the "Results of Career Planning and Educational Network Issues" section of the experimental results. By surveying students' satisfaction with their access to employment information, whether the path based on deep learning better meets students' needs and employment information needs could be understood. According to the experimental results, the path based on deep learning made students feel basically satisfied with the employment information provided by the school website, which showed that deep learning technology optimized the way students obtain employment information.

Reason: The path based on deep learning makes students feel basically satisfied with the employment information provided by the school, because deep learning technology can provide students with broader and more accurate employment information, thereby helping them better understand the needs and trends of the job market [37,38]. Gap: Although the deep learning-based path makes students feel that the employment information provided by the school is basically sufficient, there are still some students who feel that the information is not detailed enough because the deep learning model has not been fully utilized and students have different needs and expectations for information.

The hypothesis of H3 can be verified in the "Results of unclear career planning methods" section of the experimental results. By comparing students' understanding of career planning methods, whether the path based on deep learning can promote students' self-development and adaptability can be understood. According to the experimental results, the path based on deep learning enables students to have a deeper understanding of career planning methods, which shows that deep learning technology promotes students' improvement of their self-development and adaptability.

Reason: The path based on deep learning enables students to have a deeper understanding of career planning methods, because deep learning technology can provide rich information and fully meet students' needs in information-based career planning and employment strategy formulation, thereby promoting students' improvement of self-development and adaptability [39]. Gap: Despite the improved understanding, further observation and research are still needed on whether this approach can be truly applied to real life and career planning and whether it can help students better adapt to future career environments.

## Conclusions

With the advent of the big data era, it is difficult for people to leave their computers most of the time. To cultivate and educate college students' awareness of career planning, people's concepts needs to be changed. This study aims to explore the application of deep learning in career

planning and employment strategies in the information age, and compare career planning paths based on deep learning models with traditional career planning paths. It is found that the career planning path based on deep learning can better meet students' needs for career planning and employment strategy knowledge. Compared with the traditional path, the path based on deep learning enables students to have a deeper understanding of career planning and helps them develop better plans and strategies. Second, this study highlights the importance of universities in shaping and promoting students' awareness of career planning. Colleges should not only provide necessary knowledge and skills training, but also serve as a bridge between students and the job market, helping them understand the match between their abilities and career requirements, so that their careers can be better planned. However, this study also has some limitations. First, the sample size is small, which may affect the representativeness of the results. Secondly, due to the limitations of the experimental design, there may be other influencing factors that have not been taken into account. In subsequent research, we will continue to expand the research on deep neural networks in the field of career design and explore its impact mechanism on career development.

The project combines personal career design based on deep neural networks with employment decision-making theory, which has important theoretical and practical significance for the career development of college graduates. The research aims to explore the career planning of college students based on deep learning, and lay a theoretical foundation for the formulation and implementation of career planning and implementation plans for college graduates. In addition, this article will also provide an innovative platform for the research and practice of career planning in colleges and universities, so that it can better adapt to the career development needs of college students. In the long run, students, employers and the entire education system can benefit from this data-based career planning. Based on the above research results, this article will explore the specific application of deep learning in various industries, and explore its relationship with the current teaching policy and actual situation.

## Supporting information

**S1 Checklist.**
(PDF)

## Author Contributions

**Writing – original draft:** Yichi Zhang.

**Writing – review & editing:** Yichi Zhang.

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
