## [Decision Letter · Decision Letter 0]

2 Feb 2024

PONE-D-23-39644Path of Career Planning and Employment Strategy Based on Deep Learning in the Information AgePLOS ONE

Dear Dr. Zhang,

Thank you for submitting your manuscript to PLOS ONE. After careful consideration, we feel that it has merit but does not fully meet PLOS ONE’s publication criteria as it currently stands. Therefore, we invite you to submit a revised version of the manuscript that addresses the points raised during the review process.

**ACADEMIC EDITOR: **Please address all comments raised by the reviewers. ==============================

We look forward to receiving your revised manuscript.

Kind regards,

Kashif Ali, PH.D

Academic Editor

PLOS ONE

Journal Requirements:

3. For studies reporting research involving human participants, PLOS ONE requires authors to confirm that this specific study was reviewed and approved by an institutional review board (ethics committee) before the study began. Please provide the specific name of the ethics committee/IRB that approved your study, or explain why you did not seek approval in this case.

3. In the online submission form you indicate that your data is not available for proprietary reasons and have provided a contact point for accessing this data. Please note that your current contact point is a co-author on this manuscript. According to our Data Policy, the contact point must not be an author on the manuscript and must be an institutional contact, ideally not an individual. Please revise your data statement to a non-author institutional point of contact, such as a data access or ethics committee, and send this to us via return email. Please also include contact information for the third party organization, and please include the full citation of where the data can be found.

Reviewers' comments:

Reviewer's Responses to Questions

**Comments to the Author**

1. Is the manuscript technically sound, and do the data support the conclusions?

Reviewer #1: Partly

Reviewer #2: Partly

Reviewer #3: No

2. Has the statistical analysis been performed appropriately and rigorously? 

Reviewer #1: No

Reviewer #2: Yes

Reviewer #3: No

3. Have the authors made all data underlying the findings in their manuscript fully available?

Reviewer #1: No

Reviewer #2: No

Reviewer #3: No

4. Is the manuscript presented in an intelligible fashion and written in standard English?

Reviewer #1: No

Reviewer #2: Yes

Reviewer #3: No

5. Review Comments to the Author

Reviewer #1: The study idea is novel and holds potential for publication; however, there are several points that need consideration before acceptance. Firstly, the author should adhere to a standard format for the study, incorporating rationale, research questions based on identified gaps, and the study structure in the introduction section. Secondly, in the literature review, the author should formulate hypotheses based on the identified gaps. Thirdly, the methodology section appears significantly weak. The researcher has chosen an “experimental design” without proper justification or literature support. It is crucial for the researcher to explain the chosen experimental design, providing a detailed process and supporting literature. This should include information on how the experiments were conducted, control variables, and the overall methodology. Lastly, the article lacks clarity on how the data were analyzed and what statistical tests were employed before reaching a conclusion. In summary, substantial improvements are needed for the article to meet the standards for acceptance.

Reviewer #2: In the first paragraph of introduction well explained but there is a lack of literature support and evidence.

Include recent literature evidence.

Well explained the concepts career planning, deep learning and employment strategy but in this study there is a need to explore the relationship between these variables.

Need to explain what is an employment strategy and its significance in career planning.

Explain clearly how deep learning helps to plan career with literature evidence and include more empirical evidence.

Where is an objective of the study?

Provide clarity on methodology applied in the study i.e which method used or applied to analyse the data and how you choose respondents which method selected to select respondents.

Mention the role of universities to mould career and employment strategies which boosts the study.

Missing clarity on relation between career planning and deep learning. Provide clarity on it. Explain how deep learning techniques helps students to plan their career.

How this study correlates job hopping with deep learning as job hopping relates individual psychological state of taking decision on job where as deep learning is a software tool.

Career planning is a process of planning one’s career. How deep learning will act as a suitable tool to create employment to students according to their preferences.

Already there exist a number of employment online platforms integrating employees and employers. What is the strategy that distinct other online platforms from that of the proposed website in the study.

Reviewer #3: I appreciate the opportunity to review this manuscript, which addresses a topic pertinent to modern educational research. However, there are several areas where the manuscript does not meet the scholarly standards expected of academic papers.

1Introduction

The current introduction lacks essential content. While it outlines the background of deep learning, it fails to provide a clear definition, identify the research gap, or articulate the novelty of the study. The introduction should also offer an explanation of the research methodology and data sources used. Given the focus on university students’ career planning and employment strategy, a rationale for selecting this specific group for study should be included.

I suggest that the authors begin the introduction with a definition of deep learning.

Literature Support

The article lacks sufficient literature support. For example, the following statement is missing corresponding references:

"Tavabie J A studied career planning for the non-clinical workforce, that is, opportunities to develop a sustainable workforce in primary care.

Research showed that the path of career planning and employment strategy based on deep learning in the information age can better help unemployed people understand career planning and formulate employment strategies to finally choose a correct job."

2 Career Planning and Employment Strategies in the Information Age of Deep Learning

This section is intended to summarize existing literature on deep learning in career planning, but it does not appear to serve this purpose.

Additionally, a significant portion of this section is dedicated to explaining the Deep Learning Mode, which seems only tangentially related to the main topic.

3 Experiment Design and Validation

The research methods described in this section lack theoretical grounding, and there is insufficient detail regarding the acquisition of data from the two schools, observation periods, and whether the experiments were conducted by the researchers or used publicly available data.

The manuscript does not provide details about the survey content or the number of responses collected.

If the two schools differ significantly in size or have not been adequately exposed to deep learning, the comparative results lack credibility.

4Conclusions

The conclusions are overly simplistic and do not provide an adequate summary of the research or a comparison with existing studies.

6. PLOS authors have the option to publish the peer review history of their article (what does this mean?). If published, this will include your full peer review and any attached files.

Reviewer #1: No

Reviewer #2: **Yes: **Dr Prabir Chandra Padhy

Reviewer #3: No

---

## [Author Response · Author response to Decision Letter 0]

27 Feb 2024

5. Review Comments to the Author

Reviewer #1: The study idea is novel and holds potential for publication; however, there are several points that need consideration before acceptance. Firstly, the author should adhere to a standard format for the study, incorporating rationale, research questions based on identified gaps, and the study structure in the introduction section. 

Answer:Based on your suggestions, the basic principles, research questions based on the discovered gaps, and research structure have been added to the introduction to ensure the completeness of the article.

Secondly, in the literature review, the author should formulate hypotheses based on the identified gaps. 

Answer:Based on your opinions, after reviewing the literature, hypotheses are proposed and explained based on the gaps found, giving readers something to think about.

Thirdly, the methodology section appears significantly weak. The researcher has chosen an “experimental design” without proper justification or literature support. It is crucial for the researcher to explain the chosen experimental design, providing a detailed process and supporting literature. This should include information on how the experiments were conducted, control variables, and the overall methodology. 

Answer:Based on your suggestions, an explanation of selected experimental designs has been added to the Experimental section, providing detailed procedures, including information on how the experiments were performed, control variables, and overall methodology, for easy reference by readers.

Lastly, the article lacks clarity on how the data were analyzed and what statistical tests were employed before reaching a conclusion. In summary, substantial improvements are needed for the article to meet the standards for acceptance.

Answer:Thank you for your comments. Now I would like to provide a detailed explanation of how the data was analyzed in the article and what statistical tests were used before drawing conclusions.

Reviewer #2: In the first paragraph of introduction well explained but there is a lack of literature support and evidence.

Answer:Based on your suggestions, I added literature support to the first paragraph of the introduction to improve the persuasiveness of the article.

Include recent literature evidence.

Answer:Based on your suggestion, citations for 2023 have been added to the literature.

Well explained the concepts career planning, deep learning and employment strategy but in this study there is a need to explore the relationship between these variables.

Answer:Based on your suggestions, Chapter 2 has been added to explore the relationship between variables such as career planning, deep learning and employment strategies to make it easier for readers to understand.

Need to explain what is an employment strategy and its significance in career planning.

Answer:Based on your opinions, the introduction of employment strategies and their significance in career planning has been supplemented in Section 2.2.

Explain clearly how deep learning helps to plan career with literature evidence and include more empirical evidence.

Answer：Based on your opinion, clearly explain how deep learning can help in career planning through literature evidence and including more empirical evidence in Section 2.4.1.

Where is an objective of the study?

Answer:Thank you for your opinion. The purpose of the research is introduced in the last paragraph of Chapter 1.

Provide clarity on methodology applied in the study i.e which method used or applied to analyse the data and how you choose respondents which method selected to select respondents.

Answer:Based on your suggestion, the methods applied in the study are clearly stated in Section 3.1, i.e. which method is used or applied to analyze the data, and an introduction on how to select the respondents and which method to select the respondents.

Mention the role of universities to mould career and employment strategies which boosts the study.

Answer:Thanks for your suggestions, the role of universities in shaping careers and employment strategies that promote research is detailed in Chapter 2 for the convenience of the reader.

Missing clarity on relation between career planning and deep learning. Provide clarity on it. Explain how deep learning techniques helps students to plan their career.

Answer:Provide a full explanation of the relationship between career planning and deep learning based on your recommendations. And introduce how deep learning technology can help students plan their careers.

How this study correlates job hopping with deep learning as job hopping relates individual psychological state of taking decision on job where as deep learning is a software tool.

Answer:Thank you for your suggestion. In Section 2.1, we will explain how job hopping is related to deep learning.

Career planning is a process of planning one’s career. How deep learning will act as a suitable tool to create employment to students according to their preferences.

Answer:Based on your suggestion, section 2.4.2 has been added with an introduction to how deep learning will serve as a suitable tool to create employment opportunities for students based on their preferences.

Already there exist a number of employment online platforms integrating employees and employers. What is the strategy that distinct other online platforms from that of the proposed website in the study.

Answer:Based on your suggestions, in Chapter 2, the strategies to distinguish some existing online employment platforms that integrate employees and employers from the website proposed in the study are introduced in detail to facilitate readers' comparative analysis.

Reviewer #3: I appreciate the opportunity to review this manuscript, which addresses a topic pertinent to modern educational research. However, there are several areas where the manuscript does not meet the scholarly standards expected of academic papers.

1Introduction

The current introduction lacks essential content. While it outlines the background of deep learning, it fails to provide a clear definition, identify the research gap, or articulate the novelty of the study. The introduction should also offer an explanation of the research methodology and data sources used. Given the focus on university students’ career planning and employment strategy, a rationale for selecting this specific group for study should be included.

Answer:Thank you for your comments in providing a clear definition of deep learning in the introduction and clarifying the novelty of the research.

I suggest that the authors begin the introduction with a definition of deep learning.

Answer:Based on your suggestions, I will introduce the definition of deep learning to facilitate readers' understanding of technical methods.

Literature Support

The article lacks sufficient literature support. For example, the following statement is missing corresponding references:

"Tavabie J A studied career planning for the non-clinical workforce, that is, opportunities to develop a sustainable workforce in primary care.

Answer:Thank you for your opinion. Tavabie J A's research corresponds to the literature [2], which is convenient for the author to search.

Research showed that the path of career planning and employment strategy based on deep learning in the information age can better help unemployed people understand career planning and formulate employment strategies to finally choose a correct job."

Answer:Thank you for your opinion. Based on this article, we have added relevant literature references to provide readers with an accurate basis for argumentation.

2 Career Planning and Employment Strategies in the Information Age of Deep Learning

This section is intended to summarize existing literature on deep learning in career planning, but it does not appear to serve this purpose.

Answer:Based on your suggestions, Chapter 2 of deep learning on career planning and employment strategies in the information age has been expanded and the latest literature has been introduced to achieve the purpose of research summary.

Additionally, a significant portion of this section is dedicated to explaining the Deep Learning Mode, which seems only tangentially related to the main topic.

Answer:According to your suggestion, there is indeed too much introduction to the deep learning mode part, and now some parts have been deleted to increase the focus on the topic of this article.

3 Experiment Design and Validation

The research methods described in this section lack theoretical grounding, and there is insufficient detail regarding the acquisition of data from the two schools, observation periods, and whether the experiments were conducted by the researchers or used publicly available data.

Answer:Thank you for your suggestion. The research method does lack a theoretical foundation. Now the experimental design has been supplemented with a detailed introduction to the data acquisition, observation period, experimental details, etc. of the two schools.

The manuscript does not provide details about the survey content or the number of responses collected.

If the two schools differ significantly in size or have not been adequately exposed to deep learning, the comparative results lack credibility.

Answer:Based on your comments, details regarding the survey content and number of responses collected are now provided, and the two schools are of similar size.

4Conclusions

The conclusions are overly simplistic and do not provide an adequate summary of the research or a comparison with existing studies.

Answer:Thank you for your suggestions. The conclusion has been rewritten, adding a full summary of the research and comparison with existing research to ensure the completeness of the conclusion.

---

## [Decision Letter · Decision Letter 1]

5 Apr 2024

PONE-D-23-39644R1Path of Career Planning and Employment Strategy Based on Deep Learning in the Information AgePLOS ONE

Dear Dr. Zhang,

Thank you for submitting your manuscript to PLOS ONE. After careful consideration, we feel that it has merit but does not fully meet PLOS ONE’s publication criteria as it currently stands. Therefore, we invite you to submit a revised version of the manuscript that addresses the points raised during the review process.

We look forward to receiving your revised manuscript.

Kind regards,

Kashif Ali, PH.D

Academic Editor

PLOS ONE

Journal Requirements:

Additional Editor Comments:==============================

**ACADEMIC EDITOR: **Please address all comments and suggestions raised by the reviewers.==============================

Reviewers' comments:

Reviewer's Responses to Questions

**Comments to the Author**

1. If the authors have adequately addressed your comments raised in a previous round of review and you feel that this manuscript is now acceptable for publication, you may indicate that here to bypass the “Comments to the Author” section, enter your conflict of interest statement in the “Confidential to Editor” section, and submit your "Accept" recommendation.

Reviewer #1: (No Response)

Reviewer #2: All comments have been addressed

Reviewer #3: (No Response)

2. Is the manuscript technically sound, and do the data support the conclusions?

Reviewer #1: Yes

Reviewer #2: Partly

Reviewer #3: No

3. Has the statistical analysis been performed appropriately and rigorously? 

Reviewer #1: Yes

Reviewer #2: No

Reviewer #3: No

4. Have the authors made all data underlying the findings in their manuscript fully available?

Reviewer #1: Yes

Reviewer #2: Yes

Reviewer #3: No

5. Is the manuscript presented in an intelligible fashion and written in standard English?

Reviewer #1: No

Reviewer #2: No

Reviewer #3: No

6. Review Comments to the Author

Reviewer #1: Dear author,

I would like to express my appreciation for your dedication and hard work in addressing the queries raised during the peer review. While the quality of your manuscript has improved, I still believe there are a few points that must be incorporated before the article can be accepted for publication in the PLOS One journal.

Firstly, the introduction should encompass background information, the research problem, proposed solutions, study objectives, and support from prior literature in a concise and synthesized manner. Although you have made improvements, the article remains quite lengthy. It would be beneficial to transfer additional conceptual details to the literature review section.

Secondly, in the literature review, you have proposed hypotheses, but they are not presented in a standard format. I would suggest consulting relevant articles or books on how to develop hypotheses effectively.

Thirdly, it is essential to develop hypotheses and articulate each one at the end of its corresponding relationship, along with justifications and identified gaps. This can be incorporated into the literature review section or presented under a new heading, such as "Development of Hypotheses."

Fourthly, the frequent use of the word "will" has been noted throughout the manuscript. To enhance clarity and readability, it is advisable to avoid repetitive use of "will," considering that the study has already been conducted and submitted for publication. Additionally, seeking the assistance of a professional language editor may further improve the article's language quality and overall coherence.

Reviewer #2: (No Response)

Reviewer #3: Thank you for the revisions submitted to your manuscript. However, I have a few concerns that I believe need to be further clarified to enhance the manuscript's quality.

Response to Reviewers' Comments:

I find the responses to the reviewers' comments somewhat unclear. There appears to be a lack of specific, targeted explanations for the points raised by each reviewer. Instead, the responses seem to be generalized. It would be beneficial for the manuscript if you could provide more detailed, point-by-point responses to each of the reviewers' comments, clarifying how you have addressed their individual concerns.

Literature References:

Regarding the addition of literature references, there are two main issues that I have identified:

The specific references that have been added have not been communicated to us. It would be helpful to have a list of the new references included in the revision.

Upon reviewing the revised manuscript, I noticed that only two references are highlighted in red. Given the complexity of the paper, I am concerned whether the total of 24 references cited is sufficient to support the argumentation. Please consider whether additional relevant literature should be included to adequately frame your research within the current body of knowledge.

Methodology and Relevance to Career Planning:

The manuscript provides a complex introduction to the deep learning model but does not clearly articulate its relevance to Career Planning. If one of the schools in the comparison is using deep learning, it is important to discuss how this relates to the methodological approach described. Furthermore, a more detailed explanation of why this model was chosen and its significance to the study's objectives would be beneficial.

Data Availability and Analysis:

I note that the questionnaire and data used in the manuscript have not been provided. According to PLOS ONE's requirements, the availability of raw data is a critical aspect of the submission. Please ensure that the original dataset is made accessible in accordance with the journal's policies.

However, it is possible that my system does not show the raw data and the content of the questionnaire.

Additionally, I have concerns about the data analysis. The figures presented suggest a simple statistical approach; however, it is unclear if this approach is sufficient for the complexity of the questions being addressed. Please provide more information on the data analysis methods used and consider whether a more sophisticated analysis might better support the findings.

questionnaire and Response Rate:

You have now provided the details regarding the s questionnaire and the number of responses collected, which is appreciated. However, it remains unclear how the size similarities between the two schools might impact the study's findings or the generalizability of the results. A brief discussion on this matter would be valuable

7. PLOS authors have the option to publish the peer review history of their article (what does this mean?). If published, this will include your full peer review and any attached files.

Reviewer #1: No

Reviewer #2: No

Reviewer #3: No

---

## [Author Response · Author response to Decision Letter 1]

30 May 2024

Reviewer #1: Dear author,

I would like to express my appreciation for your dedication and hard work in addressing the queries raised during the peer review. While the quality of your manuscript has improved, I still believe there are a few points that must be incorporated before the article can be accepted for publication in the PLOS One journal.

Firstly, the introduction should encompass background information, the research problem, proposed solutions, study objectives, and support from prior literature in a concise and synthesized manner. Although you have made improvements, the article remains quite lengthy. It would be beneficial to transfer additional conceptual details to the literature review section.

Answer: Based on your opinion, the introduction needs to be shortened to a certain extent. The additional conceptual details of deep learning has been transferred to the method section, because if they are included in the literature review section, it will result in the introduction being too long.

Secondly, in the literature review, you have proposed hypotheses, but they are not presented in a standard format. I would suggest consulting relevant articles or books on how to develop hypotheses effectively.

Answer: Based on your opinion, the hypotheses in the literature review have been presented in a standard format.

Thirdly, it is essential to develop hypotheses and articulate each one at the end of its corresponding relationship, along with justifications and identified gaps. This can be incorporated into the literature review section or presented under a new heading, such as "Development of Hypotheses." "

Answer: According to your request, an introduction to each hypothesis is added at the end of the hypothesis's corresponding relationship, and an additional chapter is set up as "Development of Hypotheses" to clarify the reasons and identify gaps for the convenience of readers.

Fourthly, the frequent use of the word "will" has been noted throughout the manuscript. To enhance clarity and readability, it is advisable to avoid repetitive use of "will," considering that the study has already been conducted and submitted for publication. Additionally , seeking the assistance of a professional language editor may further improve the article's language quality and overall coherence.

Answer: The use of the word "will" has been avoided, and the paper's language quality and overall coherence have been improved.

Reviewer #2: (No Response)

Reviewer #3: Thank you for the revisions submitted to your manuscript. However, I have a few concerns that I believe need to be further clarified to enhance the manuscript's quality.

Response to Reviewers' Comments:

I find the responses to the reviewers' comments somewhat unclear. There appears to be a lack of specific, targeted explanations for the points raised by each reviewer. Instead, the responses seem to be generalized. It would be beneficial for the manuscript if you could provide more detailed, point-by-point responses to each of the reviewers' comments, clarifying how you have addressed their individual concerns.

Literature References:

Regarding the addition of literature references, there are two main issues that I have identified:

The specific references that have been added have not been communicated to us. It would be helpful to have a list of the new references included in the revision.

Answer: Thank you for your opinion. The new references included in the revision have been listed.

Upon reviewing the revised manuscript, I noticed that only two references are highlighted in red. Given the complexity of the paper, I am concerned whether the total of 24 references cited is sufficient to support the argumentation. Please consider whether additional relevant literature should be included to adequately frame your research within the current body of knowledge.

Answer: Thank you for your opinion. Five more references have been supplemented to support the argument.

Methodology and Relevance to Career Planning:

The manuscript provides a complex introduction to the deep learning model but does not clearly articulate its relevance to Career Planning. If one of the schools in the comparison is using deep learning, it is important to discuss how this relates to the methodological approach described. Furthermore. , a more detailed explanation of why this model was chosen and its significance to the study's objectives would be beneficial.

Answer: Thank you for your comment. An explanation of the correlation between deep learning models and career planning in the paw has been supplemented, and a more detailed explanation of the reasons for choosing this model and its significance for research objectives has been provided.

Data Availability and Analysis:

I note that the questionnaire and data used in the manuscript have not been provided. According to PLOS ONE's requirements, the availability of raw data is a critical aspect of the submission. Please ensure that the original dataset is made accessible in accordance with the journal's policies .

However, it is possible that my system does not show the raw data and the content of the questionnaire.

Answer: Thank you for your opinion. Since this paper is based on field survey data, the original data is private and does not support public display and query.

Additionally, I have concerns about the data analysis. The figures presented suggest a simple statistical approach; however, it is unclear if this approach is sufficient for the complexity of the questions being addressed. Please provide more information on the data analysis methods used and consider whether a more sophisticated analysis might better support the findings.

Answer: Based on your suggestion, more information about the data analysis method used has been added to the experimental design section, and the feasibility of solving the complexity of the problem solved by this experimental data analysis method has been supplemented.

---

## [Decision Letter · Decision Letter 2]

18 Jun 2024

PONE-D-23-39644R2Path of Career Planning and Employment Strategy Based on Deep Learning in the Information AgePLOS ONE

Dear Dr. Zhang,

Thank you for submitting your manuscript to PLOS ONE. After careful consideration, we feel that it has merit but does not fully meet PLOS ONE’s publication criteria as it currently stands. Therefore, we invite you to submit a revised version of the manuscript that addresses the points raised during the review process.

ACADEMIC EDITOR: I have gone through the reviewer comments, few aspects are very important, hypotheses should not develop in introduction. Please read the prior literature and follow them to craft the introduction, and hypotheses development. 2nd reviewer raised major concerns related to deep learning. Please address and justify the comments.  

We look forward to receiving your revised manuscript.

Kind regards,

Kashif Ali, PH.D

Academic Editor

PLOS ONE

Reviewers' comments:

Reviewer's Responses to Questions

**Comments to the Author**

1. If the authors have adequately addressed your comments raised in a previous round of review and you feel that this manuscript is now acceptable for publication, you may indicate that here to bypass the “Comments to the Author” section, enter your conflict of interest statement in the “Confidential to Editor” section, and submit your "Accept" recommendation.

Reviewer #1: (No Response)

Reviewer #3: All comments have been addressed

2. Is the manuscript technically sound, and do the data support the conclusions?

Reviewer #1: Partly

Reviewer #3: No

3. Has the statistical analysis been performed appropriately and rigorously? 

Reviewer #1: No

Reviewer #3: No

4. Have the authors made all data underlying the findings in their manuscript fully available?

Reviewer #1: Yes

Reviewer #3: No

5. Is the manuscript presented in an intelligible fashion and written in standard English?

Reviewer #1: No

Reviewer #3: (No Response)

6. Review Comments to the Author

Reviewer #1: Though the author has made changes in response to the reviewers' queries, these changes seem superficial. For instance, the author was asked to revise the literature review section; however, the hypotheses are presented in the introduction, which is also confusing. Similarly, the author was instructed to move identical or additional information to the literature review section, but it was instead relocated to the methods section. Moreover, the article lacks a proper literature review section. Additionally, it is unclear how the author has tested the proposed hypotheses. Details about the study variables are provided, but there is no supporting literature, undermining the acceptance of these details without identifying gaps in prior research. Overall, it appears that the author did not adequately address the reviewers' suggestions, resulting in considerable confusion regarding the article's structure and organization. The author seems to need a better understanding of the fundamental components of an academic article and their purposes.

Reviewer #3: I am writing to address the issues previously raised in my review. Despite the responses provided, I remain unconvinced.

1.Data Availability Policy

According to the information available on the journal's website, it is mandatory to make the data publicly accessible:

The following policy applies to all PLOS journals, unless otherwise noted.

Introduction

PLOS journals require authors to make all data necessary to replicate their study’s findings publicly available without restriction at the time of publication. When specific legal or ethical restrictions prohibit public sharing of a data set, authors must indicate how others may obtain access to the data.

https://journals.plos.org/plosone/s/data-availability

“”author Answer: Thank you for your opinion. Since this paper is based on field survey data, the original data is private and does not support public display and query.“”

2. Inappropriate Hypothesis Integration:

Incorporating the hypothesis within the introduction section is highly unorthodox and not acceptable for a scholarly manuscript. This approach detracts from the academic rigor expected in a PLOS ONE submission.

3. Insufficient References:

The manuscript contains only 29 references, which is inadequate for the scope of the research. A comprehensive literature review is essential for establishing the validity and context of your study.

4.Misunderstanding of Deep Learning Concepts！

The most critical issue lies in your misinterpretation of the concept of "deep learning." As explained: Deep Learning in Education vs. AI: In the field of education, "deeper learning" refers to an educational approach that emphasizes students’ deep understanding of core academic content, higher-order thinking skills, and the quality of learning. It focuses on students' active engagement and the application of knowledge in real-world scenarios to prepare them for future societal challenges. In contrast, in the field of artificial intelligence, "deep learning" is a machine learning technique based on artificial neural networks, particularly those with multiple hidden layers. It involves training models using large datasets and complex algorithms to accomplish tasks such as speech recognition, image recognition, and natural language processing. Despite the similar names, they are fundamentally different:

Deeper Learning in Education: Focuses on students' deep understanding and application of knowledge, critical thinking, and problem-solving skills.

Deep Learning in AI: A machine learning technique utilizing deep neural networks to automatically learn features and patterns from data.

The manuscript erroneously applies the AI concept of deep learning to educational research without clarifying how it pertains to the educational context. Furthermore, you have not demonstrated how the two schools in your study are implementing AI-based deep learning in vocational education. Considering the current educational landscape in China, it is highly improbable that such advanced teaching methodologies are in place, and the relevant educators are likely not equipped with the necessary expertise.

Additionally the manuscript fails to provide an accurate neural network structure that could appropriately explain the research problem in the context of AI-based deep learning.

7. PLOS authors have the option to publish the peer review history of their article (what does this mean?). If published, this will include your full peer review and any attached files.

Reviewer #1: No

Reviewer #3: No

---

## [Author Response · Author response to Decision Letter 2]

25 Jun 2024

Reviewer #1: Though the author has made changes in response to the reviewers' queries, these changes seem superficial. For instance, the author was asked to revise the literature review section; however, the hypotheses are presented in the introduction, which is also confusing. Similarly, the author was instructed to move identical or additional information to the literature review section, but it was instead relocated to the methods section. Moreover, the article lacks a proper literature review section. Additionally, it is unclear how the author has tested the proposed hypotheses. Details about the study variables are provided, but there is no supporting literature, undermining the acceptance of these details without identifying gaps in prior research. Overall, it appears that the author did not adequately address the reviewers' suggestions, resulting in considerable confusion regarding the article's structure and organization. The author seems to need a better understanding of the fundamental components of an academic article and their purposes.

A: Thank you for your opinion. I have reviewed the full text of the article and modified the missing parts. I hope you can review it again. I hope my modifications can make the article more complete

Reviewer #3: I am writing to address the issues previously raised in my review. Despite the responses provided, I remain unconvinced.

A: Thank you for your opinion. I have reviewed the full text of the article and modified the missing parts. I hope you can review it again. I hope my modifications can make the article more complete

1.Data Availability Policy

According to the information available on the journal's website, it is mandatory to make the data publicly accessible:

The following policy applies to all PLOS journals, unless otherwise noted.

Introduction

PLOS journals require authors to make all data necessary to replicate their study’s findings publicly available without restriction at the time of publication. When specific legal or ethical restrictions prohibit public sharing of a data set, authors must indicate how others may obtain access to the data.

https://journals.plos.org/plosone/s/data-availability

“”author Answer: Thank you for your opinion. I have submitted the data involved to your journal. If the article is published successfully, readers can contact the corresponding author's email address if they have any questions.“”

2. Inappropriate Hypothesis Integration:

Incorporating the hypothesis within the introduction section is highly unorthodox and not acceptable for a scholarly manuscript. This approach detracts from the academic rigor expected in a PLOS ONE submission.

Answer: According to the suggestion, adjust the hypothesis content in the introduction to the theoretical part to maintain the rigor of the article.

3. Insufficient References:

The manuscript contains only 29 references, which is inadequate for the scope of the research. A comprehensive literature review is essential for establishing the validity and context of your study.

Answer:The literatures in this paper are added to 35, which increases the validity and background of the research.

4.Misunderstanding of Deep Learning Concepts！

The most critical issue lies in your misinterpretation of the concept of "deep learning." As explained: Deep Learning in Education vs. AI: In the field of education, "deeper learning" refers to an educational approach that emphasizes students’ deep understanding of core academic content, higher-order thinking skills, and the quality of learning. It focuses on students' active engagement and the application of knowledge in real-world scenarios to prepare them for future societal challenges. In contrast, in the field of artificial intelligence, "deep learning" is a machine learning technique based on artificial neural networks, particularly those with multiple hidden layers. It involves training models using large datasets and complex algorithms to accomplish tasks such as speech recognition, image recognition, and natural language processing. Despite the similar names, they are fundamentally different:

Deeper Learning in Education: Focuses on students' deep understanding and application of knowledge, critical thinking, and problem-solving skills.

Deep Learning in AI: A machine learning technique utilizing deep neural networks to automatically learn features and patterns from data.

The manuscript erroneously applies the AI concept of deep learning to educational research without clarifying how it pertains to the educational context. Furthermore, you have not demonstrated how the two schools in your study are implementing AI-based deep learning in vocational education. Considering the current educational landscape in China, it is highly improbable that such advanced teaching methodologies are in place, and the relevant educators are likely not equipped with the necessary expertise.

Additionally the manuscript fails to provide an accurate neural network structure that could appropriately explain the research problem in the context of AI-based deep learning.

Answer:In order to distinguish conceptual misunderstandings, add an explanation in the introduction to clarify that this research focuses on deep learning techniques in the field of artificial intelligence:In order to eliminate conceptual confusion, this study specifically points out that 'Deeper Learning in Education' in the field of education and 'Deep Learning in AI' in the field of artificial intelligence are two completely different concepts. This study focuses on the latter, namely, the use of deep learning techniques in artificial intelligence to enhance students' career planning and employment strategy capabilities. The implementation of this technology and its application prospects in the field of education will be described in detail below.

At the same time, add a new section detailing how to implement AI-based deep learning techniques in vocational education, including the neural network structures and algorithms used:In this study, deep learning techniques are applied to the educational field of career planning and employment strategies. The technology relies on multi-layer neural network models, including CNN and LSTM, to enable in-depth analysis of educational data. The CNN structure captures local features of image and text data through its multilayer filters, while the LSTM network optimizes the processing of sequence data to learn and predict long-term dependencies in time series. These models provide students with personalized career development advice and employment strategies by training complex patterns in learning data. In the study, the application of deep learning technology not only improves students' cognition level of career planning, but also optimizes their access to employment information, and promotes self-development and adaptability. Through empirical analysis, this study validates the effectiveness of deep learning application in the field of education and demonstrates its potential in enhancing students' career competitiveness.

---

## [Decision Letter · Decision Letter 3]

10 Jul 2024

PONE-D-23-39644R3Path of Career Planning and Employment Strategy Based on Deep Learning in the Information AgePLOS ONE

Dear Dr. Zhang,

Thank you for submitting your manuscript to PLOS ONE. After careful consideration, we feel that it has merit but does not fully meet PLOS ONE’s publication criteria as it currently stands. Therefore, we invite you to submit a revised version of the manuscript that addresses the points raised during the review process.

**ACADEMIC EDITOR: **Dear authors, address all the comments of reviewers, especially reviewer that reject your manuscript. I'm not repeating the comments again, please pay attention to address them properly. ==============================

We look forward to receiving your revised manuscript.

Kind regards,

Kashif Ali, PH.D

Academic Editor

PLOS ONE

Reviewers' comments:

Reviewer's Responses to Questions

**Comments to the Author**

1. If the authors have adequately addressed your comments raised in a previous round of review and you feel that this manuscript is now acceptable for publication, you may indicate that here to bypass the “Comments to the Author” section, enter your conflict of interest statement in the “Confidential to Editor” section, and submit your "Accept" recommendation.

Reviewer #1: (No Response)

Reviewer #4: All comments have been addressed

2. Is the manuscript technically sound, and do the data support the conclusions?

Reviewer #1: No

Reviewer #4: Yes

3. Has the statistical analysis been performed appropriately and rigorously? 

Reviewer #1: No

Reviewer #4: Yes

4. Have the authors made all data underlying the findings in their manuscript fully available?

Reviewer #1: Yes

Reviewer #4: Yes

5. Is the manuscript presented in an intelligible fashion and written in standard English?

Reviewer #1: No

Reviewer #4: Yes

6. Review Comments to the Author

Reviewer #1: I am writing to express my deep disappointment that the author has not taken the reviewers' comments seriously. This is the third revision submitted to PLOS One, yet it still fails to address the valuable feedback provided.

Peer review is a critical service to the scientific community, requiring significant time to read, analyze, and provide constructive feedback. The author must recognize that publishing in an open-access journal, particularly PLOS One, does not mean compromising on the rigor and quality of research. Adhering to journal requirements is essential for publication in such a reputable journal.

The author has evidently not invested the necessary time to incorporate the suggested changes. For example, there is still no literature review section. The paper is divided into four sections—Introduction, Theoretical Framework, Empirical Analysis, and Conclusions—but lacks a comprehensive literature review (refer to line 125, noting the absence of page numbers). The literature review could be combined with the theoretical framework.

Moreover, the three hypotheses presented (lines 606-613) are isolated and lack context, making it difficult for readers to understand their basis. Additionally, there is no section detailing the materials and methods, which hinders readers from replicating the study under similar conditions. Although the author has replied to my previous comment on how the hypotheses were tested (see lines 693-697), they did not provide their results or any further details in the subsequent sections. The absence of a discussion section persists even in this third revision, with partial discussions of results found under "Development of Hypotheses" (lines 829-878), yet these discussions lack support from prior literature. This shows that the author is still not clear about the development of hypotheses and how they can be tested.

Finally, the manuscript fails to discuss the implications of the study, its contributions to the field, and its potential beneficiaries. Overall, it appears that the author either lacks a fundamental understanding of research or is unable to incorporate the reviewers' comments effectively. I suggest that the author consult basic research methodology books or seek guidance from senior researchers to improve their research and writing skills.

Reviewer #4: All have been addressed by authors adequately. However, I suggest authors should also add one or two sentences regarding methodology in abstract.

7. PLOS authors have the option to publish the peer review history of their article (what does this mean?). If published, this will include your full peer review and any attached files.

Reviewer #1: No

Reviewer #4: No

---

## [Author Response · Author response to Decision Letter 3]

18 Jul 2024

Reviewer #1: I am writing to express my deep disappointment that the author has not taken the reviewers' comments seriously. This is the third revision submitted to PLOS One, yet it still fails to address the valuable feedback provided.

Peer review is a critical service to the scientific community, requiring significant time to read, analyze, and provide constructive feedback. The author must recognize that publishing in an open-access journal, particularly PLOS One, does not mean compromising on the rigor and quality of research. Adhering to journal requirements is essential for publication in such a reputable journal.

The author has evidently not invested the necessary time to incorporate the suggested changes. For example, there is still no literature review section. The paper is divided into four sections—Introduction, Theoretical Framework, Empirical Analysis, and Conclusions—but lacks a comprehensive literature review (refer to line 125, noting the absence of page numbers). The literature review could be combined with the theoretical framework.

Answer：The literature review part is added in the paper, which is placed between the introduction and the theory part.

Moreover, the three hypotheses presented (lines 606-613) are isolated and lack context, making it difficult for readers to understand their basis. 

Answer：The following sections have been added to the text to enhance coherence and provide context for the hypothesis：In view of the advantages of using deep learning technology for personalized career recommendation proposed in this study, this paper further discusses the specific application and potential impact of this technology on career planning path construction. Based on the fact that websites can provide more accurate analysis of users' career interests, skills, and experiences, this paper proposes the following hypotheses to evaluate the effectiveness of deep learning models in improving the cognitive depth and accuracy of college students' career development. These assumptions are not only based on the description of the technical functions of the website, but also rooted in a deep understanding of the needs of students in the education information age.

Additionally, there is no section detailing the materials and methods, which hinders readers from replicating the study under similar conditions. Although the author has replied to my previous comment on how the hypotheses were tested (see lines 693-697), they did not provide their results or any further details in the subsequent sections. 

Answer：Specific experimental Settings are added in the paper to help readers replicate the experiment under similar conditions：In this paper, students from two schools were surveyed to explore their main sources of knowledge about career planning and employment strategies. The questionnaire was designed through literature review and expert consultation, and 600 students were selected by random sampling to ensure the representativeness of the sample. The data collection followed ethical standards, including anonymity and informed consent. Descriptive statistical methods were used in statistical analysis to classify "yes" and "no" answers, and to present the proportion of dependence on school education and other information sources in percentage form.

The absence of a discussion section persists even in this third revision, with partial discussions of results found under "Development of Hypotheses" (lines 829-878), yet these discussions lack support from prior literature. This shows that the author is still not clear about the development of hypotheses and how they can be tested.

Answer：Some literatures are added to support the discussion.

Finally, the manuscript fails to discuss the implications of the study, its contributions to the field, and its potential beneficiaries. Overall, it appears that the author either lacks a fundamental understanding of research or is unable to incorporate the reviewers' comments effectively. I suggest that the author consult basic research methodology books or seek guidance from senior researchers to improve their research and writing skills.

Answer：In the conclusion section added the article's contribution to the field, as well as its potential beneficiaries content:This research brings new insights into the field of higher education and career development by leveraging deep learning techniques in the field of career planning and employment strategies. The research reveals the significant potential of deep learning to enhance students' cognitive level of careers and their ability to acquire employment strategies, providing empirical support for educators to design effective career development curricula and interventions. In addition, this study provides university career development centers with an innovative tool to meet the career development needs of students with a personalized and information-based approach. In the long run, students, employers and the education system as a whole will benefit from this data-driven approach to career planning. Future research can build on this to further explore the application of deep learning techniques at different levels of education and career fields, and explore how they integrate with current education policies and practices.

Reviewer #4: All have been addressed by authors adequately. However, I suggest authors should also add one or two sentences regarding methodology in abstract.

Answer：A discussion of methods is added to the abstract:This study combines quantitative and qualitative methods, collects data through questionnaires, and uses deep learning model for analysis. Control group and experimental group were set up to evaluate the effect of career planning education. Descriptive statistics and correlation analysis were used to ensure the accuracy and reliability of the results.

---

## [Decision Letter · Decision Letter 4]

29 Jul 2024

Path of Career Planning and Employment Strategy Based on Deep Learning in the Information Age

PONE-D-23-39644R4

Dear Dr. Zhang,

We’re pleased to inform you that your manuscript has been judged scientifically suitable for publication and will be formally accepted for publication once it meets all outstanding technical requirements.

Kind regards,

Kashif Ali, PH.D

Academic Editor

PLOS ONE

Additional Editor Comments (optional):

Reviewers' comments:

Reviewer's Responses to Questions

**Comments to the Author**

1. If the authors have adequately addressed your comments raised in a previous round of review and you feel that this manuscript is now acceptable for publication, you may indicate that here to bypass the “Comments to the Author” section, enter your conflict of interest statement in the “Confidential to Editor” section, and submit your "Accept" recommendation.

Reviewer #1: All comments have been addressed

2. Is the manuscript technically sound, and do the data support the conclusions?

Reviewer #1: Yes

3. Has the statistical analysis been performed appropriately and rigorously? 

Reviewer #1: Yes

4. Have the authors made all data underlying the findings in their manuscript fully available?

Reviewer #1: Yes

5. Is the manuscript presented in an intelligible fashion and written in standard English?

Reviewer #1: No

6. Review Comments to the Author

Reviewer #1: The author has addressed all the previously highlighted points, and the paper can be accepted with some suggestions without further review. First, replace the heading "Literature Review" with "Literature Review and Hypotheses Development". Second, replace the heading "Development of Hypotheses" with "Discussion on Results". Finally, please review the English language for cohesion and clarity.

7. PLOS authors have the option to publish the peer review history of their article (what does this mean?). If published, this will include your full peer review and any attached files.

Reviewer #1: No

---

## [Editor Report · Acceptance letter]

12 Aug 2024

PONE-D-23-39644R4 

PLOS ONE

Dear Dr. Zhang, 

I'm pleased to inform you that your manuscript has been deemed suitable for publication in PLOS ONE. Congratulations! Your manuscript is now being handed over to our production team.

Kind regards, 

on behalf of

Dr. Kashif Ali 

Academic Editor

PLOS ONE